# Function of Connexin-43 in Macrophages

**DOI:** 10.3390/ijms22031412

**Published:** 2021-01-30

**Authors:** Daniel Rodjakovic, Lilian Salm, Guido Beldi

**Affiliations:** 1Department for Visceral Surgery and Medicine, Bern University Hospital, University of Bern, CH-3010 Bern, Switzerland; daniel.rodjakovic@dbmr.unibe.ch (D.R.); lilian.salm@dbmr.unibe.ch (L.S.); 2Department for BioMedical Research (DBMR), Bern University Hospital, University of Bern, CH-3008 Bern, Switzerland

**Keywords:** connexin, hemichannel, gap junction, physiology, pathophysiology, mechanisms, connexin-43, Cx43, gja1, macrophage

## Abstract

Recent studies have helped to increase the understanding of the function of Connexin-43 (Cx43) in macrophages (Mφ). The various roles of Cx43 in Mφs range from migration, antigen-presentation and some forms of intercellular communication to more delicate processes, such as electrochemical support in the propagation of the heartbeat, immunomodulatory regulation in the lungs and in macrophage-differentiation. Its relevance in pathophysiology becomes evident in inflammatory bowel disease (IBD), tumours and HIV, in which aberrant functioning of Cx43 has been described. However, the involvement of Cx43 in other Mφ functions, such as phagocytosis and polarisation, and its involvement in other types of local and systemic inflammation, are still unclear and need further research.

## 1. Introduction

Macrophages (Mφ) have been studied for well over a hundred years [1]. Their function as phagocytic cells and as a first line of defense against pathogens was initially documented in historic work in bacteria [2,3]. Over the years, Mφs were found to be involved in processes such as regenerative responses [4], inflammation of systemic organs [5,6], tumour-killing [7], hypercholesterolemia [8], sepsis [9] as well as many others. These findings increasingly established Mφs as a diverse and heterogenic cell type of the immune system [10]. Subsequently, some of these newly found functions were discovered to be intertwined with the multifaceted protein connexin-43 (Cx43).

Connexins, in general, are essential gap-junction proteins, which enable intercellular communication via the transfer of ions and signaling molecules between cells, by forming hemichannels [11,12]. Six connexin proteins form these hexameric structures. Hemichannels are transported to the plasma membrane, where they can dock with other channels in closely adjacent cells, forming gap junction channels. These gap junction channels can accumulate in the thousands in the form of gap junction plaques, to exchange information between cells, until they are internalized as vesicles, which contain the junctional membrane of both adjacent cells. Hemichannels are also involved in long-range direct cell-to-cell communication via nano-tubes, paracrine communication in the form of undocked hemichannels/pores [13] and various other processes [14]. Mutations in connexins cause disorders ranging from changes in the skin, cataracts, and hearing loss to complex syndromes [15] and embryo lethal mutations [16].

Cx43, also known as GJA1, is one of these connexins. It is prominently featured in the immunological synapse [17], thus making it the prime target for investigations in immune cells, such as Mφs. In Mφs, Cx43-dependant intercellular signal transfer is involved in a plethora of physiological and pathophysiological processes, such as immunomodulation [18,19], regulation of the heartbeat [20] and purinergic signaling [9]. Therefore, Cx43 acts as a gateway to the Mφs outer-world, as much as it allows the environment to influence Mφs and the expression of Cx43 itself [9,21,22].

This review summarizes the current literature regarding the role of Cx43 in Mφs during physiological as well as pathophysiological processes. Deviations into other cell types, such as monocytes, are kept short and serve the purpose of clarification of Cx43-function in Mφs.

Lastly, it is important to note, that the investigations into Cx43 in relation to Mφs are limited, since the homozygous Cx43 knockout is lethal in mice due to heart failure [16,23]. These circumstances have led to studies making use of different circumventions, such as heterozygous mice [24,25], Mφ-Cx43 specific knockout mice [9,18] and fetal liver cell transplantation from homozygous Cx43 knockout mice into irradiated recipient-mice [26], as discussed below.

## 2. Dawdling and Devouring: Rolling around for Initiative

Mφs are essentially phagocytic cells. They migrate, phagocytose and present antigens to other cells of the immune system.

### 2.1. Migration

The expression of Cx43 in lymphocytes was proposed as early as 1997, as the potential interaction between Mφs and the vessel wall endothelium during infiltration was considered [8]. However, blocking connexins with connexin mimetic peptides had little influence on Mφ transendothelial migration [27]. The influence of Cx43 in Mφ-migration was further investigated in studies using cells from heterozygous Cx43 knockout mice [24,25]. Interestingly, newer studies present Mφ-Cx43 as not only being capable of regulating the migratory ability of Mφs, but also as influencing the migration of other cells [24,28].

Increases in Cx43 expression consequently enhances the migratory ability of Mφs [25]. Intriguingly, the expression of Cx43 in Mφs is increased in lipopolysaccharide (LPS) conditioned medium [9,22,29]. This was identified during a transwell-assay with LPS-conditioned medium, in which Cx43 expression was found to be increased. However, migration can be inhibited by immunosuppressive drugs, such as mTOR inhibitors and steroids, as well as by downregulation of Cx43 expression [25].

Moreover, the transmigratory behavior of freshly isolated human Mφs, was analyzed in a blood–brain barrier model [21]. They were pretreated with TNF-α plus IFN-γ to induce Cx43 expression. It was found that transmigration was associated with the formation of heterocellular gap junctions in-between monocytes/Mφs and endothelial cells. Blocking the gap junctions reduced the number of Mφs, which were found to have transmigrated across the model. After transmigration, Cx43 was prominently found by staining in monocytes/Mφs, endothelial cells, and astrocytes, while being more intense at heterocellular contacts [21].

Furthermore, altered secretion of signaling molecules by heterozygous Cx43 Mφs, decreased migration of other cells, such as smooth muscle cells [24]. The same was found for neutrophils, although this was not the case for Mφs from homozygous fetal livers [28]. Moreover, heterozygous Cx43 Mφs have also been identified as altering the progression of diseases, such as atherosclerosis, by influencing cell migration. Atherosclerosis progression was found to be less prominent in Cx43 heterozygous mice, as their plaques contain significantly fewer neutrophils, due to reduced Mφ-induced chemotaxis and subsequent accumulation [28]. Overall, influence over the Mφs-microenvironment is closely tied to Cx43 and will be discussed more thoroughly in chapter three through to chapter five.

### 2.2. Phagocytosis

After arriving at the inflammation site by migration or transmigration, Mφs start to phagocytose, engulfing and digesting pathogens, thus acting as first line responders of the innate immune system.

Cx43 was initially proposed to be involved in phagocytosis, as various functions, some of which are essential in phagocytosis, were found to rely upon connexins. However, this was later found to not be the case, as described in the conflicting results below.

Anand R.J. et al. found significant, although partial inhibition of phagocytosis in heterozygous Cx43 knockout mice and oleamide-inhibited murine Mφs [30]. They obtained heterozygote Cx43 knockout Mφs via peritoneal lavage from heterozygote mice and homozygous Cx43 knockout Mφs from the embryonic livers of homozygous knockout mice. The cells were identified as Mφs based upon their surface expression of CD45 [30] using coverslips [31]. The Mφs were given sheep erythrocytes, latex beads, and *Escherichia coli* to phagocytose and were then evaluated by light and confocal microscopy [30].

Glass A.M. et al. on the other hand found no difference in the phagocytic capabilities between wild type and Cx43 knockout Mφs [26]. They obtained Mφs through peritoneal lavage and bone marrow harvesting from previously irradiated chimeric mice, which were reconstituted with homozygous Cx43 knockout fetal liver cells. The cells were then identified by flow cytometry as CD11b and F4/80 double positive. Mφs were given sheep erythrocytes, zymosan particles and *Listeria monocytogenes* while their phagocytic ability was measured by fluorescent microscopy and flow cytometry. Glass A.M. et al. criticized the methodology used by Anand R.J. et al., as CD45 is widely accepted as being present in multiple cell types derived from hematopoietic cells and not just on Mφs [26], unlike F4/80, which is specific to murine Mφs [32].

Another recent study by Dosch M. et al. found phagocytosis to be independent of Cx43 [9]. The Mφs were obtained from conditional Mφ Cx43 knockout mice via peritoneal lavage. The cells were incubated with latex beads and evaluated with a phagocytosis assay kit. Neither pharmacological blocking with Gap27 nor Cx43 deletion altered phagocytosis [9].

It is unclear, if the different methodologies, i.e., derivation of Mφs and material used to phagocytose, may have impacted the Mφs ability to properly perform phagocytosis. More studies covering different methodologies are needed to further investigate target-dependent phagocytosis.

### 2.3. Antigen-Presentation

Mφs and dendritic cells (DCs) were found to cooperate in the uptake, transfer, and presentation of antigens [33]. Different possibilities for Cx43-dependent antigen transfer were discussed, until the transfer of loaded major histocompatibility complex (MHC) class II molecules was proposed to be Cx43-dependant trogocytosis, as molecules from donor cells were detected on the surface of acceptor cells [33]. Trogocytosis is a process whereby lymphocytes extract surface molecules of antigen presenting cells and express them on their own membrane [34,35,36]. MHC class I presentation on the other hand, was found to be identical between wild type and homozygous Cx43 knockout Mφs, in the case of *Listeria monocytogenes* infection [26]. Of note, Mφs with deletion of Cx43 are more proficient in T cell priming. The mechanism behind this, is postulated to be an increased accumulation of antigens, since these Mφs are unable to transfer them to neighboring DCs, which leads to efficient presentation [33].

However, antigen presentation is not the only Cx43-dependant intercellular communication in Mφs, as described in the following chapter.

## 3. Intercellular Communication Is a Two-Way Street

Intercellular communication is essential for the function of the immune system, and the established connections between Mφs and parenchymal cells exhibit Cx43 dependent functions. Cx43-dependent communication by Mφs has also been proven to be essential in non-immune functions, such as the regulation of the heartbeat [20] and diseases, such as HIV [37].

### 3.1. Physiological Communication in Heart, Lung and Intestine

#### 3.1.1. Electrochemical Communication of the Heart

Mφs are well known to influence cardiac disease and repair [38]. However, they have only recently been implicated in a nonimmune context [39]. Mφs in the AV node, as well as the left and right ventricular walls, all express Cx43. FACS-sorted cells were used to identify expression, to eliminate potential cross contamination with Cx43 high-expressing cardiomyocytes. Cx43 was then analysed in Mφs by whole-mount immunofluorescence. It was found to be marked on punctate contacts between Mφs and cardiomyocytes. Electron microscopy then confirmed direct cellular contact. This led to the conclusion, that Mφs couple to cardiomyocytes using Cx43 containing gap junctions. The Mφs do so primarily in the distal part of the AV node. Interestingly, these Mφs regulate the heartbeat by reducing the action potential and aiding in repolarisation, thus allowing higher conduction rates. This was confirmed by Mφ ablation using diphtheria toxin, which led to an AV block.

Furthermore, Mφ Cx43 is also speculated to be involved in additional abnormalities of the atria and ventricles, such as atrial fibrillation and ischemia-induced ventricular arrhythmias [20].

#### 3.1.2. Immunomodulatory Communication in the Lungs

The involvement of Cx43 in alveolar Mφs has also only recently been described [18,19]. By using real-time in situ imaging, alveolar Mφ-epithelium gap junction channels containing Cx43 were identified by Westphalen K. et al. After inducing inflammation with LPS, alveolar CD11c^cre/cre^ Cx43^floxed/floxed^ Mφs remained sessile and attached themselves to the alveoli. To investigate, if Cx43 may be responsible for the Mφ-immobility, bacteria and PBS were microinjected and the cells observed. Mφs were found to rapidly ingest the bacteria, yet they remained sessile. The study also found neutrophils freely entering and migrating ruling out non-specific physical factors. Hence, it was concluded, that Cx43 was not responsible for Mφ-immobility. Cx43 was found to play another role in these sessile Mφs. They were found to utilize the epithelium as a conducting pathway, communicating by synchronized Ca^2+^ waves. This intercellular communication was found to be immunosuppressive: activating Akt Ca2^+^-dependently [18], a serine/threonine kinase which influences cell survival, growth, proliferation, angiogenesis, metabolism, and migration [40]. Cx43 knockout alveolar Mφs were also identified increasing secretion of proinflammatory cytokines themselves (MIP-1α) and in the epithelium (CXCL1,5), indicating mutual cytokine suppression. Additionally, Cx43 knockout Mφs enhanced the alveolar recruitment of neutrophils [18].

Similar findings were also seen in human cells. Beckmann A. et al. investigated the communication between a bilayer of human Mφs and human alveolar epithelial cells. The goal was to identify if Mφ-epithelial gap junctions exist in humans. Interestingly, co-cultures were found to express Cx43, whereas isolated Mφs did not [19]. These results lay a cornerstone for future research, concerning Cx-43 dependent Mφ immunomodulation in the lungs, as well as presenting potential targets for medical treatment.

#### 3.1.3. Intercellular Communication in the Intestine

Aside from cardiomyocytes [20] and alveolar epithelium [18,19], Mφs were also found to communicate with the epithelium of the intestine. Mφs use Cx43 to form functional gap junctions with the epithelial cells and use paracrine and heterocellular signaling to communicate. This communication is directly involved in inflammatory bowel disease (IBD) (see below) [41].

### 3.2. Pathological Communication in IBD, Tumors and HIV

The use of Cx43 in intercellular communication is not exclusive to physiological processes: IBD, tumors and HIV are some of the diseases involving aberrant use of Mφ Cx43.

#### 3.2.1. IBD

Mφs establish communication with epithelial cells using Cx43, this communication then contributes to the dysregulation of the intestinal epithelial barrier in IBD [41]. Interestingly, connexin expression in IBD tissues is relocated more basolaterally in epithelial cells, compared to normal tissue. This phenomenon may facilitate the interaction of intestinal epithelial cells with infiltrating Mφs, allowing for disease progression. Intestinal epithelial cells were therefore seeded in six-well plates on top of activated Mφs to mimic the observed architecture in the intestinal tissue. Under optimal conditions, calcein dye transfer could be seen by a shift in fluorescence, demonstrating its transfer from epithelial cells to Mφs [41].

Nevertheless, such remodulation of cellular structures, as seen in IBD, is just a mild example of cellular reorganization, a more extreme example can be found in tumors.

#### 3.2.2. Filopodia in Tumor Networks and HIV

By immunostaining for Mφ markers, tumor-associated Mφs (TAM) were found in anaplastic thyroid tumors [42]. These TAMs establish a variety of intercellular contacts in-between themselves, as well as with other cancer cells. Contacts are established using long, irregular, thin, moniliform cytoplasmic processes, which express Cx43 at their tip. These cytoplasmic processes are described as nanotube-like, yet it is unclear if they are indeed nanotubes. The direct contact with other Mφs, cancer cells and blood vessels may allow communication and molecular transfer. TAMs are both evenly intermingled with cancer cells, as well as located in long chains, which derive from perivascular clusters across the tumor. These networks are robust and resilient structures, which allow for an advantage on the neighboring non-tumor tissues. By allowing intercellular passage of numerous molecules and ions, Cx43 has been shown to act as the main culprit in the coordination in intercellular signaling and efficient metabolic support in cancer cell networks. This is shown by the fact that many tumor cells were found to be more than 150 microns away from blood vessels, without any signs of necrosis or apoptosis, even though this distance is thought to represent the maximum distance of oxygen diffusion from blood vessels into the tissue [42].

Unlike the presumed nanotubes in TAMs, HIV does in fact exploit Cx43-containing tunnelling-nanotubes (TNT). In HIV-infected Mφs, Cx43 expression is induced three days post infection and remains high. The gap-junctions formed between infected and uninfected Mφs were proven to be functional by identifying Lucifer Yellow diffusion in-between these cells. No dye uptake was observed when extracellular dye was presented to infected and uninfected Mφs, suggesting that Cx43 hemichannels are only present at the tip of the TNT. The study concludes that TNTs are required for efficient intercellular communication and viral spread, hence selective Cx43 blocking may present itself as future therapeutic target against HIV [37].

## 4. A Microenvironment with Macro-Consequences

The Mφ’s interaction with the microenvironment itself provides an alternative to direct cell-to-cell contacts and serves as a much broader form of communication. Physiological and pathophysiological processes alter the Mφs environment, thus changing its expression of Cx43 [22,25,29]. The change in Cx43 expression furthermore influences the Mφs gene expression for chemokine secretion or activation of the complement pathway [28]. Moreover, Cx43 deletion also alters the release of molecules, such as adenosine triphosphate (ATP) [9].

### 4.1. The Environment Changes the Mφ: LPS, Acute Peritonitis and Sepsis

Inflammatory sites, in which Mφs aggregate, are proposed to influence connexin channels [21]. Although, LPS was once found to inhibit lymphocyte-Mφ communication via an unclear mechanism [43], some recent studies found LPS-induced Cx43 expression [22] and increase in intercellular communication, measured by dye transfer [44]. Cx43 expression in bone marrow-derived Mφs increased in a dose- and time-dependent manner during contact with LPS [22]. The same reaction was found for rat liver Kupffer cells. Cx43 was found to be predominantly localized at the cell to cell interfaces, thus indicating gap junction formation and probable intercellular communication between Kupffer cells in vivo. In vitro, Kupffer cells were in fact found to increase dye transfer after the administration of LPS and IFN-gamma, which correlates with increased Cx43 expression [44]. This may be interpreted as a higher communication rate [37,41,44]. It was also found that the LPS-dependant induction of Cx43 in Mφs via the iNOS pathway, can be significantly decreased by tacrolimus and methylprednisolone, two commonly administered immunosuppressive drugs used after solid organ transplantation [25].

Specific infectious diseases, such as the infection with *Mycobacterium tuberculosis*, can also increase expression of Cx43, thus enhancing intercellular communication, which leads to an increase in the apoptosis rate and expression of inflammatory factors [45]. It is also important to note, that murine peritoneal Mφs—which are in contact with LPS and ATP—induce other inflammatory processes, such as the activation of the NLRP3 inflammasome. Moreover, NLRP3 inflammasome activation was found to be intertwined with Cx43 expression, as heterozygous Cx43 Mφs have been shown to decrease inflammasome activation. In addition, they changed intracellular redox modulation, progressing renal inflammatory cell injury [29].

But what happens when the pathological burden of an infection overthrows the immune system? Murine models were used to assess the relation of Cx43-functionaility and mortality in sepsis and acute peritonitis. Mφ Cx43 function was found to be inversely correlated with mortality, as heterozygous Cx43 mice and pharmacologically Cx43-blocked mice show increased mortality in sepsis [30] and acute peritonitis [22]. Intriguingly, the opposite was reported in a murine sepsis model with Mφ-specific depletion of Cx43 [9]. Furthermore, Cx43 positive Mφs were found in the human peritoneal cavity of patients with peritonitis, but not in control patients, suggesting the involvement of Mφ Cx43 in septic processes in humans [9].

### 4.2. The Mφ Changes the Environment: ATP Release

Peritonitis was found to increase extracellular ATP levels, suggesting the involvement of Mφs in systematic ATP release during sepsis [9]. Extracellular ATP is utilized as autocrine regulation and paracrine communication by immune cells [46]. Active ATP-release from inflammatory cells can occur both via vesicular exocytosis and via hemichannels, including Cx43 [47,48]. The ATP-release allows for multifunctional modulation of the cell, tissue, and organism [49].

Cx43 and P2X7 purinergic receptors have been found to be co-localized on peritoneal Mφs [50]. ATP-release from Mφs occurs in response to TLR-2 and TLR-4 activation in a Cx43-dependant manner [9]. After the release, ATP typically activates P2 receptors [51,52,53,54]. These results therefore support the findings already made for the P2Y1 receptor [55,56], and other immune cells [51,57]. The findings indicate the possible involvement of Cx43-dependant ATP-release in a variety of situations, including certain inflammatory diseases [58], as well as physiological situations, i.e., intercellular immunomodulation in the lungs, as established by Westphalen K. et al. [18] (see Section 3.1.2).

## 5. Mφs Dress for the Job They Want

### 5.1. Adapting to Their Environment: Polarisation

Mφs can adapt according to changes in their microenvironment by polarization [59]. The subtypes are commonly summarized and simplified in M1 and M2 Mφs, even though polarisation is rather a spectrum than a switch [59,60,61]. The types differ in their functions:M1 Mφs accumulate at the inflammation site by strong adhesion, which promotes cell retention and the progression of inflammation [62];M2 Mφs possess stronger migratory ability compared to M1 Mφs, have increased phagocytic properties, and are in general anti-inflammatory and considered to help with tissue repair [62,63].

Morel S. et al. found that mice with Cx43-heterozygous Mφs do not differ from homozygous knockout mice or wild type Mφs in terms of M1 and M2 polarization [28]. These Mφs were derived from haematopoietic fetal liver cells, which were transferred to lethally irradiated mice [28]. The absence in change of polarization was recently confirmed for peritoneal M2 Mφs under Cx43 blocking or deletion. 

However, markers for M1 differentiation, such as iNOS and Il12RB, were found to be decreased under Cx43 blocking or deletion [9]. The diminished M1 Mφ polarization may be due to the Cx43-dependant polarisation via Angiotensin-2 (Ang-2) [64], as Ang-2 was identified as having decisive roles in inflammation: increasing adhesion molecules, cytokines, chemokines and more [65]. This may therefore influence Mφ polarisation in experiments, in which mice undergo inflammation, as is the case in the latter mentioned study [9].

The discrepancy in regard to the influence of Cx43 in Mφ polarization to the M1 type, may alternatively be due to the inherent difficulties in identifying the Mφ polarization, as markers used for identification of polarization in vitro were found to not always work in vivo. For instance, genes modulated inside the iNOS signaling pathway were found to differ slightly compared to in vitro Mφs [66].

### 5.2. Adapting to Their Environment: Differentiation

Mφs also adapt to their environment by differentiation. In two settings, differentiation was found to involve the use Cx43 in either formation or function.

#### 5.2.1. Foam Cells

Lipid-laden foam cells form when Mφs retain too much cholesterol, thus becoming the prototypical cells associated with atherosclerotic plaque [67]. Foam cells in rabbits, which had hypercholesterolemia-induced atherosclerosis, were found to express Cx43, however precursor monocytes in normocholesterolemic rabbits were not. Cx43 expression in Mφs was observed to be similar in animals which were mechanically injured by balloon injury and had hypercholesterolemia, as well as those which only had hypercholesterolemia. Moreover, neither alveolar Mφs, Kupffer cells nor Mφs from peritoneal or bronchial lavage from normo- and hypercholesterolemic rabbits were identified as inducing Cx43 expression. This contradiction of previous papers may be due to the choice of laboratory animals or methodology [8].

#### 5.2.2. Foreign Body Giant Cells

In response to biomaterials in bones, foreign body giant cells (FBGC) may form, which are derived from precursors of the monocyte/Mφ lineage [68,69]. Implanted biomaterial in minipig femura yields Cx43 positive Mφs and FBGCs. The expression was identified by immunohistochemistry, Western Blot and in situ hybridization, additionally, ultrastructural analysis of gap junctions was performed. Thus, expression of Cx43 in FGBCs and Mφs was found in the granulation tissue and on the surface of the implanted biomaterial [68]. Cx43 labelling on the protein and mRNA level was attributed to ultrastructurally identified gap junctions between Mφs, between FBGCs and between FBGCs and Mφs [68]. Furthermore, the formation of FBCG could also be shown ultrastructurally. Hence, the conclusion can be reached, that Cx43 seems to play a role in the formation of osteoclast-like FBGCs [68]. Moreover, a subsequently performed experiment, using Cx43 and microfilament labelling suggests the possible role of podosome and hemichannels in biomaterial degradation, utilizing Cx43 [69].

## 6. Conclusions

The function of Cx43 on immune cells, such as Mφs, is difficult to study, yet seems to provide useful insights to its critical importance. Only a few papers have been published, which selectively investigated floxed Cx43 Mφs. Thus, a substantial amount of the currently available research is gathered from either homozygous Mφs, which were collected from fetal organs and transplanted into a host or from heterozygous Mφs. It is unclear if the lack of Cx43 may have initiated the expression of other connexins as a compensatory mechanism, or if alternative factors influenced the sometimes-contradictory results obtained from the experiments.

Nevertheless, Cx43 and Mφ functions remain closely intertwined, as Cx43 is critical in various physiological and pathophysiological processes. This includes interactions with neighboring cells, the microenvironment, migration to inflammation sites, antigen-presentation, immunomodulation, as well as supporting macro physiological functions such as heartbeat-regulation. Diseases like IBD, HIV, sepsis and other infections all depend on—and are modulated by—Cx43. Therefore, it is likely that Cx43 is also of importance in Mφ function in other types of diseases, that remain to be explored in future studies. The research on the intricate and versatile functions of Cx43 in Mφs would allow for the establishment of specific drug targeting and modulation of the innate and adaptive immune system, thus contributing to current fields of interest for Mφ research, such as polarization, intercellular communication, inflammation, wound healing and more.

A summary of some select Cx43 functions in Mφs can be found in Figure 1. The figure depicts mechanisms, which show its diverse roles as a pore in the micro-environment, as well as in direct cell-to-cell contacts, in lung, heart and disease.

## Figures and Tables

**Figure 1 ijms-22-01412-f001:**
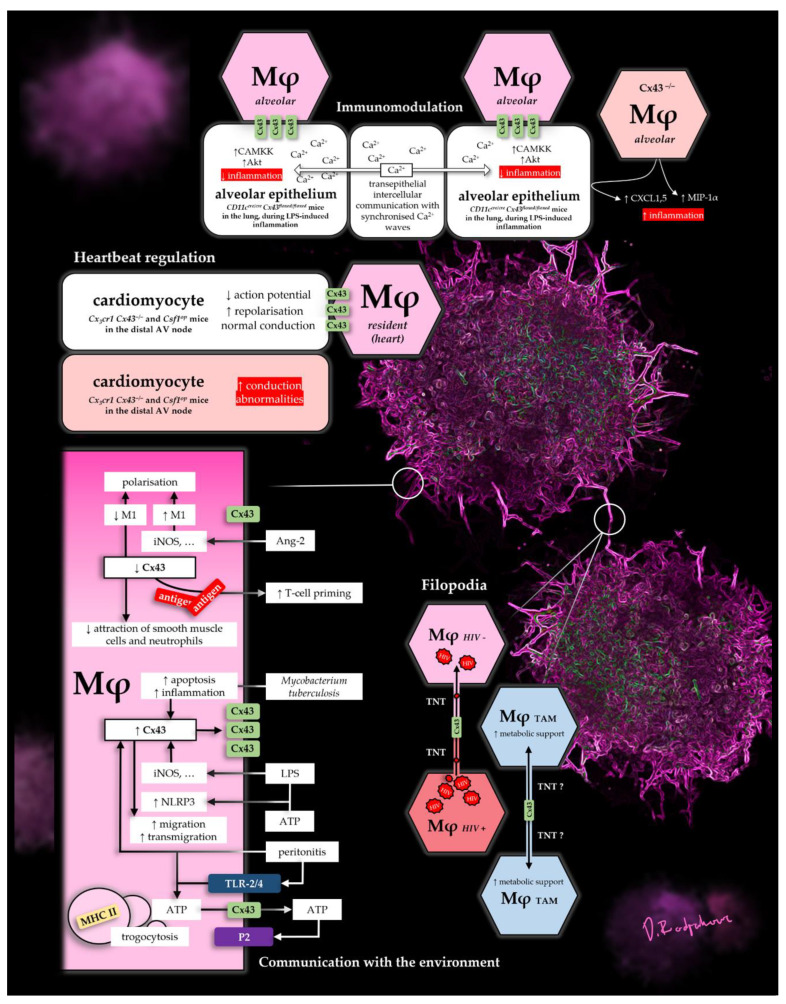
Selection of Cx43-dependant Mφ functions.

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
