# Peer review of "Function of Connexin-43 in Macrophages"

_ijms, 2021, doi:10.3390/ijms22031412_

Round 1

Reviewer 1 Report

 This review highlights many of the emerging biological roles of Cx43 in macrophages that extend beyond its role in cellular communication. These unique roles have not been highlighted well in the past so there is a nice niche for this review as it should be well received and cited. Here the authors present a unique focus on Cx43 importance on macrophage function. Overall, this reviewer thoroughly enjoyed reading this manuscript and found the information included informative, well organized and nicely presented.

As mentioned in the title, the review is mainly focused in Connexin43, however some other Cxs could be lightly mentioned through the text as a point of comparison to Cx43  and to reinforce the uniqueness of Cx43 role on this system, per example role of macrophage Cx43 vs Cx37 in the development of atherosclerosis. It should also be considered/discussed if discrepant results observed when comparing Cx43 heterozygous vs Knockout could be explained by increased expression of a different Cx in a compensatory mechanism.   

This review would benefit from a summarizing table of functions and systems/pathological processes associated with Cx43 presence in macrophages, as well as, schematic figure(s) illustrating processes described along the review which should include details for the exact roles of Cx43 and the other factors involved, clearly highlighting the importance of this protein.

On page 5, section LPS and Cx43, the phrase:

  • “It was also found that the induction…”, seems incomplete.
  • “this model, NLRP3 inflammasome…” should be completed: “In this model,…”

On page 7:

  • Section about Foam cells is not very clear, please consider rephrasing
  • Section about Foreign body giant cells would benefit from a more complete description of why/what are the evidences showing that Cx43 plays a role on the formation of osteoclast-like FBGCs and in biomaterial degradation

It would also be helpful to include the most important research questions related to connexins and macrophages in order to get the field forward.

Overall, after editing, this review will be a nice addition to the field and will also be of interest to the greater cell biology community.

Reviewer 2 Report

The manuscript from Rodjakovic et al. presents an interesting and well integrative perspective about the role of Cx43 channels in macrophages, which might represent relevant therapeutic targets for different pathologies. It describes, in an unusual but very attractive manner, the mechanism and biological processes in which macrophage Cx43 is involved. Moreover, the way different apparently contradictory theories are presented is original and highlights the importance of controversy for knowledge progress.

This review is grounded in very recent references and papers published in reputed journal, which demonstrates the pertinence and timeliness of the topic.

Although the review is very well structured, some minor issues should be addressed in order to improve the quality of the manuscript and make it more attractive for a wider audience.

- a more detailed information concerning basic aspects of Cx43 biology should be provided

- the text requires a proof-reading since some sentences need to be rephrased to become clearer; for example “Interestingly, the newer studies present macrophage-Cx43 not only regulating the migratory ability of macrophages, but also their influencing the migration of other cells” should be replaced by “Interestingly, the newer studies present macrophage-Cx43 not only regulating the migratory ability of macrophages, but also influencing the migration of other cells”. These sentence “Although, migration can be inhibited by immunosuppressive drugs, such as mTOR inhibitors and steroids, as well as by downregulation of Cx43 expression.” is not clear. The sentence “Cx43 was initially proposed to be involved in phagocytosis, as various functions, some of which are important for phagocytosis, were found to rely on connexins, based on the conflicting results shown below.” Is confusing. These are just some examples. The manuscript should be subjected to a thorough review process by a native speaker in order to make it easier to read and follow. Some sentences are difficult to understand.

- the title of section 3.1 should be replaced by “Pathophysiological communication in heart, lung and intestine” since in addition to physiological roles of macrophages Cx43 also its involvement in heart, lung and intestine pathologies is described.

- following the comment above, the section 3.1.3 and 3.2.1. should be merged

- The section 3.2.2 and 3.2.3 should make part of a broader section entitled “Pathological communication in cancer and infection”, since more literature is available concerning the role of Cx43 in monocytes/ macrophages in these two pathological conditions.

- Once the abbreviation IBD, for inflammatory bowel disease is defined, it should be used always along the text.

- to maintain some consistence and coherence, the term Cx43 should not be included in the title of sections 4.1.1 and 4.1.2. It could be considered to merge these two sections.

- a scheme that summarizes some of the most important aspects of Cx43 in macrophages should be included.  

Reviewer 3 Report

In their review, Rodjakovic et al. attempted to synthesize the function of Cx43 in monocytes/macrophages. This is a timely idea; however the manuscript is superficial in its present form, its structure is chaotic and the information incomplete. Some examples are listed below:

- the Authors should consider a more thorough summary of the monocyte/macrophage functions and of the differences between monocytes and macrophages (also with regard to Cx43 function);

- following this point: the clarity of the paper suffers from the lack of paragraph that would describe/summarize channel-independent and channel-dependent functions of Cx43;

- accordingly, the functions of Cx43 in monocyte/macrophage migration and transmigration should be discriminated;

- the structure and function of immunological synapse should be outlined along with its interrelations with Cx43. Furthermore, an outline of the mechanisms/significance of cardiac conduction/alveolar macrophages would add to the message of chapter 3.1;

- GJIC-mediated tumor networks deserve much more attention. This point is partly addressed in 5.1. but with rather faint relationship to 3.2.3;

- The message of 3.2.3. is completely incomprehensible; etc., etc.;

Round 2

Reviewer 3 Report

No more comments

Author Response

Format and minor grammatical changes have been made to all pages, including the graphical abstract.